# Metagenomic analysis of the dynamical conversion of photosynthetic bacterial communities in different crop fields over different growth periods

**Ju-E Cheng**[1,2☯]**, Pin Su**[1☯]**, Zhan-Hong Zhang**[3]**, Li-Min Zheng**[2]**, Zhong-Yong Wang**[2]**, Muhammad Rizwan Hamid**[2]**, Jian-Ping Dai**[2]**, Xiao-Hua Du**[2]**, Li-Jie Chen**[2]**, Zhong-Ying Zhai**[2]**, Xiao-Ting Kong**[2,4]**, Yong Liu**[2]**, De-Yong Zhang**[🆔][5]*

**1** College of Plant Protection, Hunan Agricultural University, Changsha, China, **2** Hunan Plant Protection Institute, Hunan Academy of Agricultural Science, Changsha, China, **3** Hunan Vegetable Institute, Hunan Academy of Agricultural Sciences, Changsha, China, **4** Long Ping Branch, Graduate School of Hunan University, Changsha, China, **5** Hunan Hybrid Rice Research Center, Changsha, China

☯ These authors contributed equally to this work.

\* dyzhang78@163.com

**Data Availability Statement:** all fastq files are available from the NCBI database (bacterial sequences SRA: SRP193589; BioProject:

## Abstract

Photosynthetic bacteria are beneficial to plants, but knowledge of photosynthetic bacterial community dynamics in field crops during different growth stages is scarce. The factors controlling the changes in the photosynthetic bacterial community during plant growth require further investigation. In this study, 35 microbial community samples were collected from the seedling, flowering, and mature stages of tomato, cucumber, and soybean plants. 35 microbial community samples were assessed using Illumina sequencing of the photosynthetic reaction center subunit M (pufM) gene. The results revealed significant alpha diversity and community structure differences among the three crops at the different growth stages. *Proteobacteria* was the dominant bacterial phylum, and *Methylobacterium*, *Roseateles*, and *Thiorhodococcus* were the dominant genera at all growth stages. PCoA revealed clear differences in the structure of the microbial populations isolated from leaf samples collected from different crops at different growth stages. In addition, a dissimilarity test revealed significant differences in the photosynthetic bacterial community among crops and growth stages (P<0.05). The photosynthetic bacterial communities changed during crop growth. OTUs assigned to *Methylobacterium* were present in varying abundances among different sample types, which we speculated was related to the function of different *Methylobacterium* species in promoting plant growth development and enhancing plant photosynthetic efficiency. In conclusion, the dynamics observed in this study provide new research ideas for the detailed assessments of the relationship between photosynthetic bacteria and different growth stages of plants.

PRJNA533201; BioSample: SAMN11444424-SAMN11444390; URL: https://www.ncbi.nlm.nih.gov/bioproject/533201).

**Funding:** This research was supported by the National Key R&D Program of China (2017YFD0200400), the National Science Foundation of China (31701764), the 13th five-year national key research and development plan (2017YFD0101906), Hunan agricultural science and technology innovation fund (2017GC04), Hunan Natural Science Foundation (2017JJ3169), the Hunan Natural Science Foundation (2020JJ4412), the China Agriculture Research System of MOF, and MARA (CARS-16-E17)and the Hunan Provincial key research and Development Plan (2016NK2199).

**Competing interests:** The authors have declared that no competing interests exist.

## Introduction

The plant phyllosphere is a complex habitat that provides a suitable environment for various bacteria, yeast, and fungi. The bacteria in the phyllosphere can promote plant growth and protect the plants from pathogens [1]. Bacteria are the most numerous colonizers in the phyllosphere (approximately $10^7$ cells per cm$^2$) [2]. Various bacterial taxa are commonly found in this habitat, and some studies indicated that α- and γ-*Proteobacteria* and *Bacteroidetes* are the dominant bacterial colonists of the phyllosphere [1, 3–6]. Photosynthetic bacteria are present in the phyllosphere, where they positively impact plant growth and disease control [7–9]. Indeed, photosynthetic phyllosphere bacteria play important roles in nitrogen and carbon dioxide fixation and desulfurization [9–11]. In addition, these bacteria produce a variety of chemicals that can induce systemic resistance in the plant hosts [9–11]. These bacteria are common microorganisms in nature but are mostly found in marine ecosystems [12, 13], river estuaries [14], freshwater lakes [15], saline lakes [16], and soil crusts [17, 18].

Photosynthetic bacteria conduct photosynthesis through photosynthetic reaction centers containing bacterial chlorophyll [19]. The most common photosynthetic bacteria used in agricultural applications are *Rhodopseudomonas palustris*, *Methylobacterium* spp., and *Sphingomonas* spp. [9, 10, 20, 21]. In a previous study by the authors' group, *R. palustris* improved plant growth and development, enhanced plant resistance against biotic and abiotic stresses, and improved soil fertility [10]. Various *Methylobacterium* isolates can enhance seed germination [22] and inhibit plant pathogens, improving plant health [23]. *Sphingomonas* spp. are capable of suppressing disease development and inhibiting pathogen growth [24]. *Methylobacterium* spp., the first reported plant photosynthetic bacteria, we found to have important functions on plant leaf surfaces [25]. The genera *Methylobacterium* and *Sphingomonas* are consistently detected in different plant species and on different plant organs [26], especially in the plant phyllosphere, implying their intimate relationship with plants and their significance in plant health. In tomato, *R. palustris* [27], *Methylobacterium* spp. [28], and *Sphingomonas* spp. promote plant growth [29]. The same has been observed in cucumber and soybean [30–32]. Regardless, it is reasonable to believe that many photosynthetic bacteria and their community structure have yet to be discovered.

The photosynthetic reaction center subunit M (pufM) gene located in the puf operon can be used to identify anoxygenic photosynthetic bacteria in environmental samples [13, 33]. Furthermore, compared to 16S rRNA gene sequences, distinct pufM gene region sequences include more phylogenetic diversity and resolve anoxygenic photosynthetic bacteria community composition in the phyllosphere with a better resolution than 16S RNA sequencing or culture [12, 34].

We hypothesized that the photosynthetic bacteria are common in the phyllosphere and that the growth period of crops will affect the composition of the photosynthetic bacteria communities. Therefore, in this study, we selected representative crops from *Solanaceae* (tomato), *Cucurbitaceae* (cucumber), and *Leguminosae* (soybean) to study the distribution of the photosynthetic bacterial communities in different crops and at different growth stages using Illumina sequencing. The results presented in this paper help to identify the dynamic changes in photosynthetic bacterial communities during crop growth and development and increase the application potential of photosynthetic bacteria in agricultural production.

## Materials and methods

### Experimental design and sample collection

A total of 35 samples were collected from leaves of three crops at different growth stages for subsequent microbe isolation and high throughput sequencing process. The experiment was

conducted from May to August 2018 at the experimental farm of the Hunan Academy of Agricultural Sciences, Changsha, Hunan Province, China (28.22˚N, 113.26˚E). Tomato cv. Zuan-hong-meili (XinShu Seed Co., Ltd., Beijing, China), cucumber cv. Shuyan 10 (XinShu Seed Co., Ltd., Beijing, China), and soybean cv. Su-qing III (Nanjing Ideal Seedings Co., Ltd., Nanjing, China) were selected for the study. First, a large piece was randomly picked from the experimental farm. Then it was divided into nine small blocks. The size of each block was approximately 40–50 m$^2$, and the nutrients were consistent (the organic matter content of soil was 1.92%, total nitrogen 0.078%, whole phosphorus 0.023%, whole potassium 2.32%, quick acting phosphorus 1.4 mg/kg and fast acting potassium 93 mg/kg). Then, 300 plants were planted in each block of cucumber, tomato, and soybean separately. Each crop was planted in three randomly selected blocks on May 15, 2018, and sampled at the seedling (June 15, 2018), flowering (July 15, 2018), and mature (August 25, 2018) growth stages. At each growth stage, fully expanded leaves were collected from six randomly selected plants per crop per block. The leaves collected from a specific block were pooled to produce three biological replicates per crop at a specific growth stage. Each collected leaf sample was approximately 10 g and stored at 4˚C until further use. Phyllosphere microorganisms were collected from each leaf sample as previously described [35], with specific modifications. Briefly, each collected leaf sample was placed inside a 500-mL sterile conical flask containing 200 mL of 0.1 M phosphate-buffered saline (PBS). The flask was shaken at 150 rpm for 10 min at 28˚C and sonicated for 20 min using an Ultrasonic Cleaning Machine (Ningbo Scientz Biotechnology Co., Ltd., Ningbo, China). The solution was transferred into four 50-mL centrifuge tubes and centrifuged for 10 min at 12,000 rpm. The resulting phyllosphere microorganism pellets were collected and stored at -80˚C until DNA extraction.

## DNA extraction and purification

The DNA of the phyllosphere microorganisms was extracted using the MP FastDNA® SPIN Kit for Soil (MP Biochemicals, Solon, OH, USA). The PufM gene forward primer PufMF (5′-TAC GGS AAC CTG TWC TAC-3′) and reverse primer PufM-WAW (5′-AYN GCR AAC CAC CAN GCC CA-3′) were constructed according to earlier reports [12, 36]. Each primer was added with a unique 6-nt barcode. The DNA isolated from the individual samples was used for PCR amplification under the following conditions: initial denaturation at 94˚C for 5 min, followed by 35 cycles of 94˚C for 30 s, 60˚C for 30 s, and 72˚C for 30 s. The final extension step was 10 min at 72˚C. Each PCR reaction was 50 μL in volume. The quality of the PCR products was checked using 2% agarose gel by electrophoresis. The correctly sized PCR products were purified using a Novogene Gel Extraction Kit (Novogene Bioinformatics Technology Co., Ltd., Beijing, China). Equal amounts of purified PCR products representing the three crops at three different growth stages were mixed and sequenced by Novogene Bioinformatics Technology Co. Ltd. (Beijing, China) on an Illumina HiSeq X Ten (2 × 250 bp paired-end reads), according to the literature [12, 19, 37].

## Processing and analysis of sequence data

Paired-end reads were assigned to samples based on their unique barcode and truncated by cutting off the barcode and primer sequence. The paired-end reads were combined using the FLASH software [38]. Quality filtering of the raw tags was performed under specific filtering conditions to obtain high-quality clean tags [39] according to the QIIME 1.7.0 software [40] quality control process. The UPARSE software [41] was used to cluster all the effective tags to generate an operational taxonomic unit (OTU) table based on 97% sequence similarity. The photosynthetic bacteria database constructed by Novogene Bioinformatics Technology Co.,

Ltd. (Beijing, China) was used to annotate the tags at different taxonomic levels. The data were normalized based on the sample with 68,550 sequences to generate an OTU table. Alpha and beta diversity analyses were performed using the normalized data.

## Statistical analysis

Statistical differences in alpha diversity indices (i.e., the Shannon index, the inverse (Inv-) Simpson index, observed richness, Chao's estimated richness (chao1), and the relative abundance of the different taxonomic groups) were calculated using QIIME (version 1.7.0). The alpha diversity indices were analyzed using one-way analysis of variance (ANOVA) followed by Duncan's multiple range test (P<0.05) in SPSS 21.0 (IBM, Armonk, NY, USA). The tests for microbial community composition dissimilarity between pairs of groups were performed using nonparametric multi-response permutation procedures (MRPPs), analysis of similarities (ANOSIM), and nonparametric permutational multivariate ANOVA with the adonis function (Adonis) [42, 43] in R (version 2.15.3). Principal coordinate analysis (PCoA) was performed in R (version 2.15.3). A t-test was performed to identify OTUs with significant variation between groups (P<0.05) using R. Annotation of Prokaryotic Taxa (FAPROTAX) was used to predict the ecological functions of photosynthetic bacterial communities [44, 45].

## Results

### The alpha diversity of the photosynthetic bacterial communities changes with the growth stage

In this study, a total of 3,150,689 quality-filtered reads representing the photosynthetic bacterial communities in different crops were obtained through high-throughput sequencing. These sequences represented 35 leaf samples collected from three different crops at three different growth stages (quality filtering is shown in S1 Table). One sample (cucumber seedling sample) had to be removed due to the low quality of the sequences. Following the initial analysis, 6725 OTUs were found to share a 97% sequence similarity. According to the rarefaction curves, most obtained samples reached a sufficient sequencing depth (Fig 1A).

The alpha diversity indices are shown in Fig 1B–1E and Table 1, including observed species, chao1, Shannon, and Inv-Simpson. The greater the alpha diversity index, the higher the diversity of the photosynthetic bacteria in the samples. For the alpha diversity indices of the tomato phyllosphere photosynthetic bacteria, all alpha diversity indices increased after the seedling stage and then decreased following the flowering stage. The highest diversity of phyllosphere photosynthetic bacteria was observed in the flowering period. For the alpha diversity indices of the cucumber phyllosphere photosynthetic bacteria, the Shannon index decreased from the seedling stage until the mature stage, while the Inv-Simpson index, the observed richness, and Chao1 increased in the seedling stage and then decreased after the flowering stage, the highest diversity of phyllosphere photosynthetic bacteria was in the flowering period. For the alpha diversity indices of the soybean phyllosphere photosynthetic bacteria, all assessed diversity indices decreased from the seedling stage until the mature stage. The highest diversity of phyllosphere photosynthetic bacteria was in the seedling period.

### The photosynthetic bacterial communities are dissimilar among crop species and growth stages

The weighted UniFrac distance was used to calculate the pairwise distances among the phyllosphere photosynthetic bacterial communities on tomato, cucumber, and soybean at different growth stages (seedling, flowering, and maturing stages). The two axes of the PCoA explained

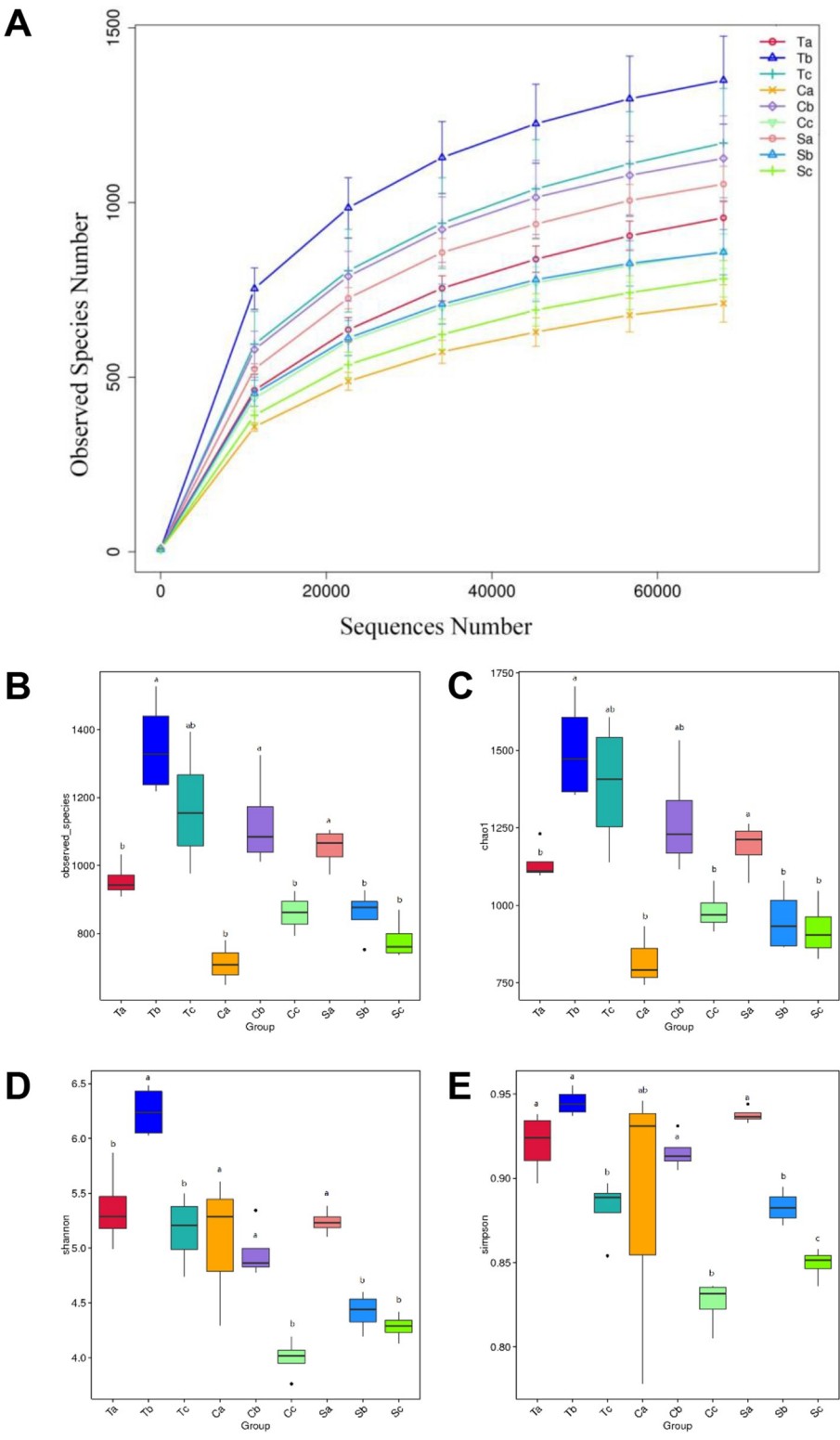

**Fig 1.** (A) Rarefaction curves of the reaction center protein M chain (pufM) gene fragment amplicons. Photosynthetic bacteria were isolated from tomato, cucumber, and soybean plants at different growth stages. The isolated photosynthetic bacterial samples were combined and sequenced with the Illumina HiSeq X Ten platform. (B-E) Alpha diversity of the three crops at three different growth stages. The alpha diversity indices include (B) the observed species, (C) Chao1, (D) Shannon index, and (E) Inv-Simpson index of photosynthetic bacterial communities. The data

were analyzed using a one-way ANOVA followed by Duncan's multiple range test at P<0.05. [a,b,c] Groups with different superscript letters are significantly different. [a]P = 0.001; [b]P = 0.008; [c]P = 0.049. n = 4. Ta, tomato seedling stage; Tb, tomato flowering stage; Tc, tomato maturity stage. Ca, cucumber seedling stage; Cb, cucumber flowering stage; Cc, cucumber maturity stage. Sa, soybean seedling stage; Sb, soybean flowering stage; Sc, soybean maturity stage.

61.76% and 14.07% of the total variation, respectively (Fig 2). The ANOSIM, MRPP, and Adonis test results showed that all variables (i.e., different growth stages of tomato, cucumber, and soybean) examined in this study explained the variation in photosynthetic bacterial community structure among tomato, cucumber, and soybean at different growth stages (Table 2). The MRPP, ANOSIM, and Adonis results also suggested that the differences in the phyllosphere photosynthetic bacterial community among different crop species at different growth stages were significant (P<0.05).

## The photosynthetic bacterial communities were different among crops and growth stages

*Proteobacteria* were the major photosynthetic bacteria in the phyllosphere. The phylum accounted for 50.21%-96.87% of the sequences obtained from the three crops at the three growth stages (Fig 3A). The abundances of photosynthetic bacterial genera found in this study were *Methylobacterium* (25.56%-94.07%), *Halorhodospira* (0.15%-22.31%), *Rhodopseudomonas* (0.06%-8.85%), *Ectothiorhodospira* (0.18%-20.51%), *Sphingomonas* (0.001%-5.26%), *Thiorhodococcus* (0.22%-5.33%), *Rhodobacter* (0.09%-3.78%), *Rhodocyclus* (0%-5.92%), *Roseateles* (0.05%-2.78%), and *Marichromatium* (0.03%-1.41%) (Fig 3B).

At the species level, photosynthetic bacteria were also found in the phyllosphere and included *Methylobacterium extorquens* (15.80%-75.99%), *Methylobacterium radiotolerans* (5.74%-65.94%), *Halorhodospira halophila* (0.04%-28.50%), *Rhodopseudomonas palustris* (0.06%-8.85%), *Ectoth- iorhodospira shaposhnikovii* (0.18%-20.51%), *Sphingomonas ursincola* (0–8.33%), *Thiorhodococcus minor* (0.17%-5.21%), *Rhodobacter veldkampii* (0.05%-3.32%), and *Rhodocyclus tenuis* (0–5.92%), and *Roseateles depolymerans* (0.06%-2.78%) (0.02%-3.67%) (Fig 3C). At this level, there were significant differences among crops at three different growth stages.

**Table 1. Alpha diversity of three vegetable crops at three different growth stages.**

| Sample group | | Observed species | Shannon | Inv-Simpson | Chao1 |
|---|---|---|---|---|---|
| Tomato | Ta | 956.25±26.75[b] | 5.36±0.19[b] | 0.92±0.01[a] | 1136.96±31.52[b] |
| | Tb | 1350.25±72.74[a] | 6.25±0.12[a] | 0.95±0[a] | 1501.3±84.28[a] |
| | Tc | 1170.25±90.22[ab] | 5.16±0.17[b] | 0.88±0.01[b] | 1389.63±106.74[ab] |
| Cucumber | Ca | 711.33±37.88[b] | 5.06±0.4[a] | 0.89±0.05[ab] | 821.97±56.61[b] |
| | Cb | 1126.5±70.07[a] | 4.96±0.13[a] | 0.92±0.01[a] | 1277.17±91.16[a] |
| | Cc | 860.25±28.61[b] | 4±0.09[b] | 0.83±0.01[b] | 983.46±34.7[b] |
| Soybean | Sa | 1052.75±29.49[a] | 5.24±0.06[a] | 0.94±0[a] | 1190.22±41.69[a] |
| | Sb | 857.75±37.23[b] | 4.42±0.09[b] | 0.88±0.01[b] | 952.48±51.9[b] |
| | Sc | 781.75±30.27[b] | 4.28±0.06[b] | 0.85±0[c] | 920.75±47.46[b] |

The alpha diversity indices include the observed species, Chao1, inverse (Inv-) Simpson, and Shannon indices of photosynthetic bacterial communities. The data were analyzed using a one-way ANOVA followed by Duncan's multiple range test at P<0.05. [a,b,c] Groups with different superscript letters are significantly different (P<0.05). Ta, tomato seedling stage; Tb, tomato flowering stage; Tc, tomato maturity stage. Ca, cucumber seedling stage; Cb, cucumber flowering stage; Cc, cucumber maturity stage. Sa, soybean seedling stage; Sb, soybean flowering stage; Sc, soybean maturity stage.

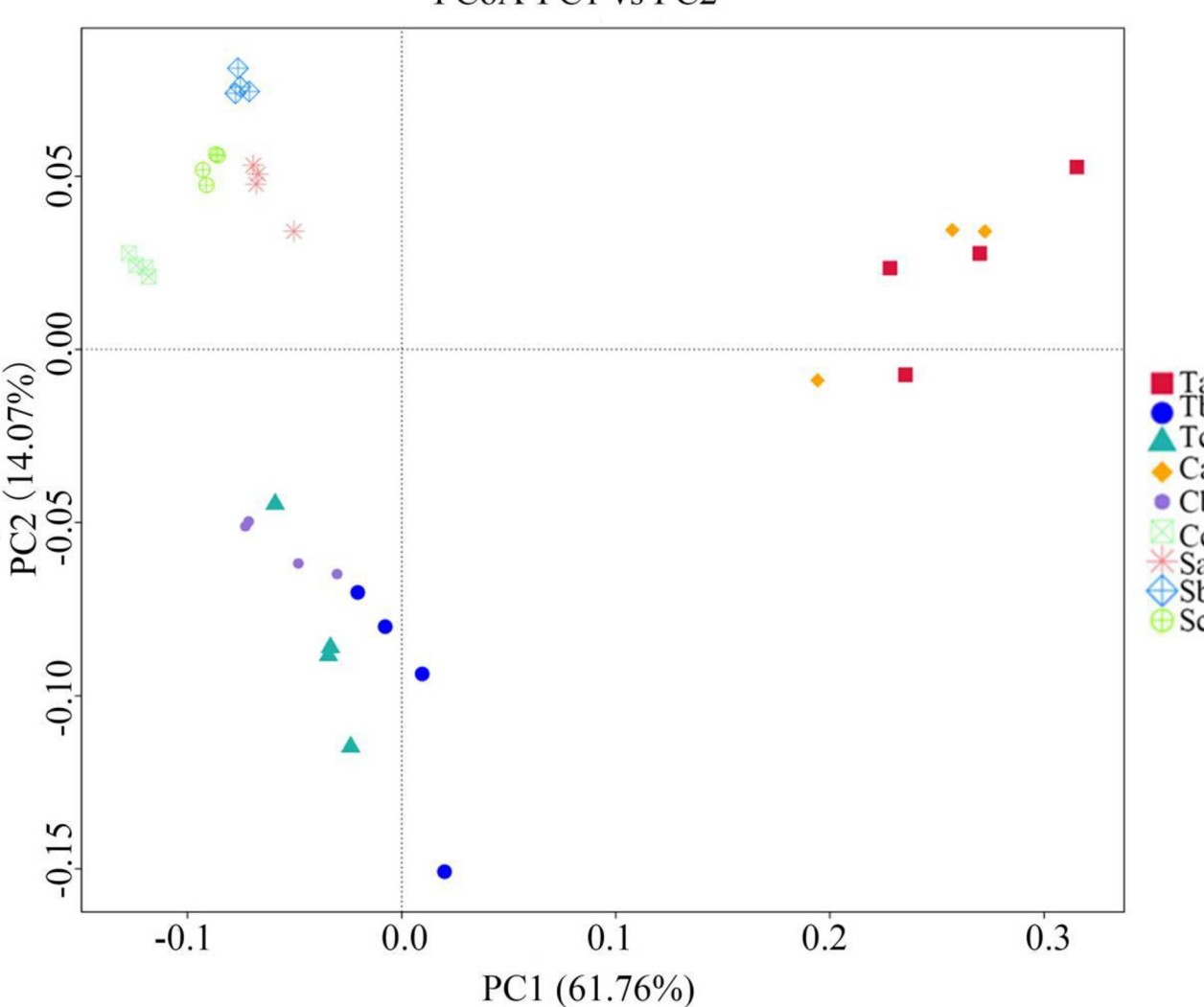

**Fig 2. Principal coordinate analysis (PCoA) of photosynthetic bacterial communities.** Each point represents one sample, plotted by a principal component on the X-axis and another on the Y-axis, which is colored by group. The percentage on each axis indicates its contribution to the discrepancy among samples. Ta, tomato seedling stage; Tb, tomato flowering stage; Tc, tomato maturity stage. Ca, cucumber seedling stage; Cb, cucumber flowering stage; Cc, cucumber maturity stage. Sa, soybean seedling stage; Sb, soybean flowering stage; Sc, soybean maturity stage.

### The photosynthetic bacteria populations change with growth stages in the three crops

The species showing significant differences at the three growth stages are shown in Figs 4–6. The bacterial strains showing a relative abundance of over 1% were selected and analyzed. During tomato growth, *M. extorquens* and *M. radiotolerans* increased in the seedling stage and then decreased after the flowering stage. *R. depolymerans* decreased from the seedling stage until the mature stage. *Sphingomonas ursincola* and *T. minor* decreased beginning in the seedling stage and then remained low after that (Table 3).

During cucumber growth, *Ectothiorhodospira shaposhnikovii* and *M. extorquens* increased after the seedling stage and then decreased after the flowering stage. *Halorhodospira halophila*, *Rhodobacter veldkampii*, *Rhodopseudomonas palustris*, and *R. depolymerans* decreased

**Table 2. Beta diversity of three different vegetable crops at three different growth stages.**

| Group | MRPP | | Anosim | | Adonis | |
|---|---|---|---|---|---|---|
| | delta | P | R | P | F | P |
| Ta-Tb | 0.3642 | 0.033 | 1 | 0.031 | 15.97 | 0.025 |
| Ta-Tc | 0.3326 | 0.033 | 1 | 0.023 | 18.864 | 0.001 |
| Tb-Tc | 0.2493 | 0.024 | 1 | 0.041 | 13.129 | 0.025 |
| Ca-Cb | 0.3068 | 0.035 | 1 | 0.024 | 19.417 | 0.001 |
| Ca-Cc | 0.2893 | 0.034 | 1 | 0.022 | 20.756 | 0.04 |
| Cb-Cc | 0.1472 | 0.03 | 1 | 0.044 | 71.614 | 0.033 |
| Sa-Sb | 0.1072 | 0.026 | 1 | 0.025 | 93.007 | 0.041 |
| Sa-Sc | 0.1229 | 0.032 | 1 | 0.037 | 102.85 | 0.035 |
| Sb-Sc | 0.1078 | 0.025 | 1 | 0.026 | 86.631 | 0.001 |

MRPP, Multi-response permutation procedure analysis; Anosim, Analysis of similarity; Adonis, A method of nonparametric multivariate variance test according to the distance matrix. $P < 0.05$ indicated a statistically significant difference.

Ta, tomato seedling stage; Tb, tomato flowering stage; Tc, tomato maturity stage. Ca, cucumber seedling stage; Cb, cucumber flowering stage; Cc, cucumber maturity stage. Sa, soybean seedling stage; Sb, soybean flowering stage; Sc, soybean maturity stage.

beginning in the seedling stage and decreased again after the flowering stage. *M. radiotolerans* increased beginning in the seedling stage and increased again after the flowering stage. *S. ursincola* decreased beginning in the seedling stage and was maintained at a low level until the maturity stage (Table 4).

During soybean growth, *E. shaposhnikovii* decreased beginning in the seedling stage and then increased after the flowering stage until the maturity stage. *M. extorquens* first increased beginning in the seedling stage and then decreased after the flowering stage. *M. radiotolerans* increased beginning in the seedling stage and increased again after the flowering stage (Table 5).

## Predicted photosynthetic bacteria community ecological functions using FAPROTAX

The predicted ecological functions of the phyllosphere photosynthetic bacteria communities were investigated using FAPROTAX (Fig 7). Among the putative functions, a total of 33 putative biogeochemical cycle functions were identified from the phyllosphere photosynthetic bacteria community. The results indicated that the photosynthetic bacteria community contained a high number of sequences assigned to phototrophy (14.4%), photoheterotrophy (14.0%), aerobic anoxygenic phototrophy (11.7%), chemoheterotrophy (11.9%), ureolysis (10.9%), methanol oxidation (10.9%), and methylotrophy (10.9%). In addition, many sequences were predicted to have ecological functions involved in photoautotrophy (3.2%), anoxygenic photoautotrophy (3.2%), anoxygenic photoautotrophy S oxidizing (3%), anoxygenic photoautotrophy $H_2$ oxidizing (1.4%), and aerobic chemoheterotrophy (1%). Other ecological functions, including nitrogen fixation, nitrate reduction, dark iron oxidation, and sulfide oxidation, were also predicted to be present in the phyllosphere photosynthetic bacterial communities in relatively lower abundances (<1%).

## Discussion

Numerous studies investigated the colonization, species, and abundance of plant phyllosphere microorganisms, but most studies focused on the roles of phyllosphere microorganisms in

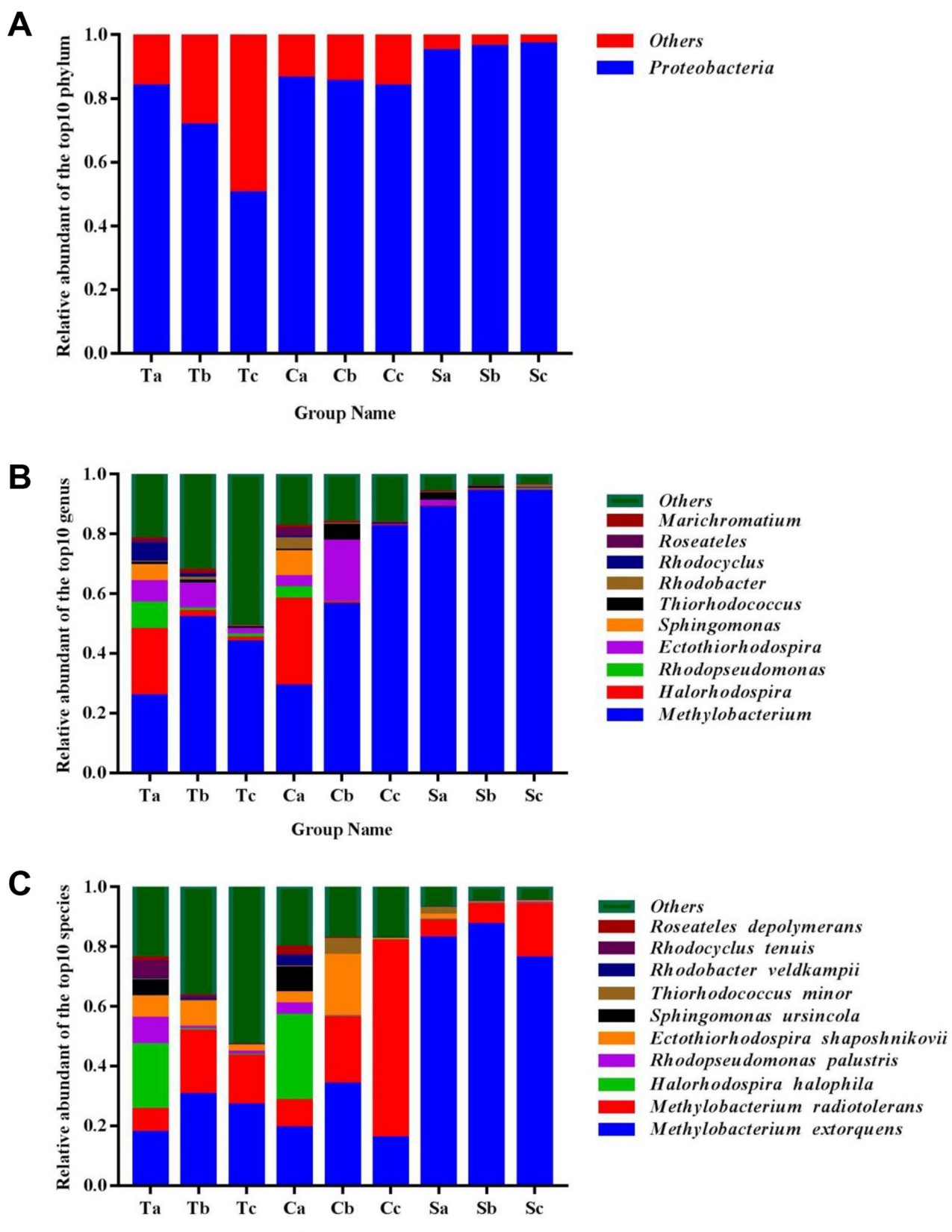

**Fig 3. Relative abundance of photosynthetic bacteria in tomato, cucumber, and soybean.** (A) Phyla. (B) Genera. (C) Species. Ta, tomato seedling stage; Tb, tomato flowering stage; Tc, tomato maturity stage. Ca, cucumber seedling stage; Cb, cucumber flowering stage; Cc, cucumber maturity stage. Sa, soybean seedling stage; Sb, soybean flowering stage; Sc, soybean maturity stage.

plant growth, development, and disease resistance [46–49]. The results showed that the photosynthetic bacterial community was common in the phyllosphere, and the composition of the photosynthetic bacterial community at the phylum and genus levels were similar among the growth stages of the three crops. In addition, the dissimilarity test revealed a significant difference in photosynthetic bacterial community composition among the crops at different growth stages.

The alpha diversity results revealed that the phyllosphere photosynthetic bacterial communities of cucumber and soybean decreased as the season progressed, while the phyllosphere photosynthetic bacterial community of tomato increased from the seedling stage to the flowering stage and then decreased after the flowering stage. Copeland et al. [37] found that the leaf microbiome had the highest diversity at the beginning of the growing season and then became significantly less diverse as the season progressed. Redford et al. [50] reported that the composition of the phyllosphere bacterial community varied greatly throughout the growing season, and the diversity of early- and late-season communities was greater than that of mid-season communities. The results for the cucumber and soybean phyllosphere photosynthetic bacteria obtained here agreed with those of Copeland et al. [37] but were inconsistent with the results of Redford et al. [50]. Copeland et al. [37] reported that the soil microbiome strongly influenced the leaf microbiome's diversity in common beans and soybean at the beginning of the growing season. In contrast, Redford et al. [50] reported that diversity was the lowest at the mid-season, probably due to adverse growth conditions (e.g., UV exposure, moisture conditions, resource availability, or leaf cuticle properties) and affecting phyllosphere microbiome colonization. There have been no studies on the phyllosphere microbial changes in cucumber, and the soybean material examined in this study was the same as that examined by Copeland et al. [37]. The photosynthetic bacterial community changes in cucumber and soybean might also be due to soil microorganisms' influence on the phyllosphere of crop seedlings. In addition, the results for tomatoes were inconsistent with those reported by Copeland et al. [37] and Redford et al. [50], as the highest community diversity of phyllosphere photosynthetic bacteria was observed during tomato flowering. A possible reason for the discrepancy is that the colonization of phyllosphere microorganisms varies among crops. Additional studies examining various crops cultured under the same conditions and analyzed using the same methods within the same study are necessary to compare the phyllosphere diversity among crops.

Indeed, there were significant differences in photosynthetic bacterial community composition among the crops and growth stages in the present study. The results of the PCoA showed clear differences among the three crops and the three growth stages. In addition, the dissimilarity test revealed significant differences in photosynthetic bacterial community composition among the three crops and three growth stages. A previous study reported that bacterial communities were plant species-specific [5]. Redford et al. [50] showed that the phyllosphere bacterial community's composition was significantly affected by the growing season. In the present study, the colonization of plant leaf surfaces by photosynthetic bacteria might also have been affected by crop species, plant growth, and development. Potential functions of the photosynthetic bacteria community determined by FAPROTAX included phototrophy, photoheterotrophy, aerobic anoxygenic phototrophy, chemoheterotrophy, ureolysis, methanol oxidation, and methylotrophy, which were abundant among all samples. For the three crops, the functional diversity of photosynthetic bacteria at the seedling stage was higher than in other growth

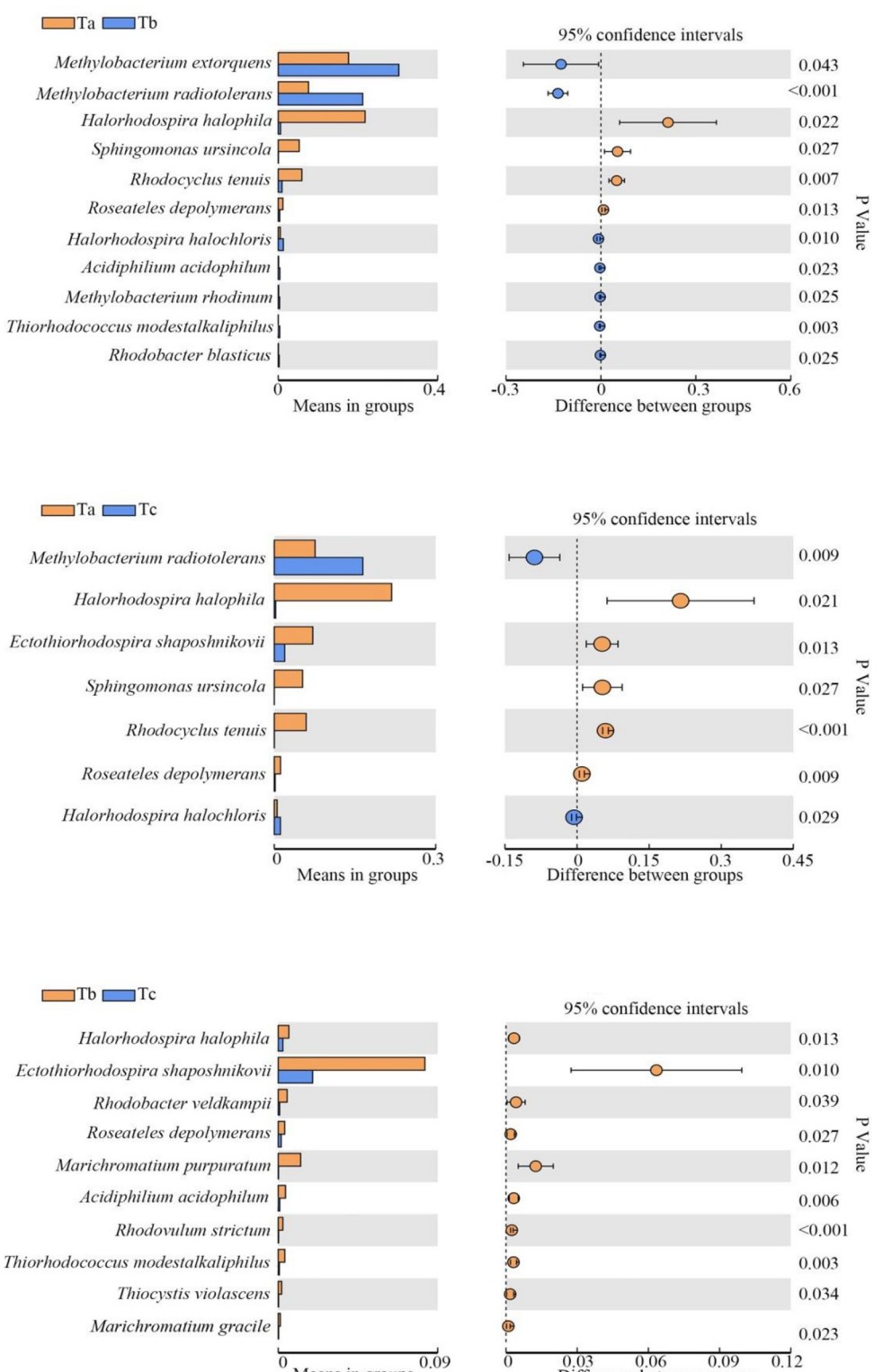

**Fig 4. Photosynthetic bacterial populations showing significant changes among tomato samples collected at different growth stages.** Ta, tomato seedling stage; Tb, tomato flowering stage; Tc, tomato maturity stage.

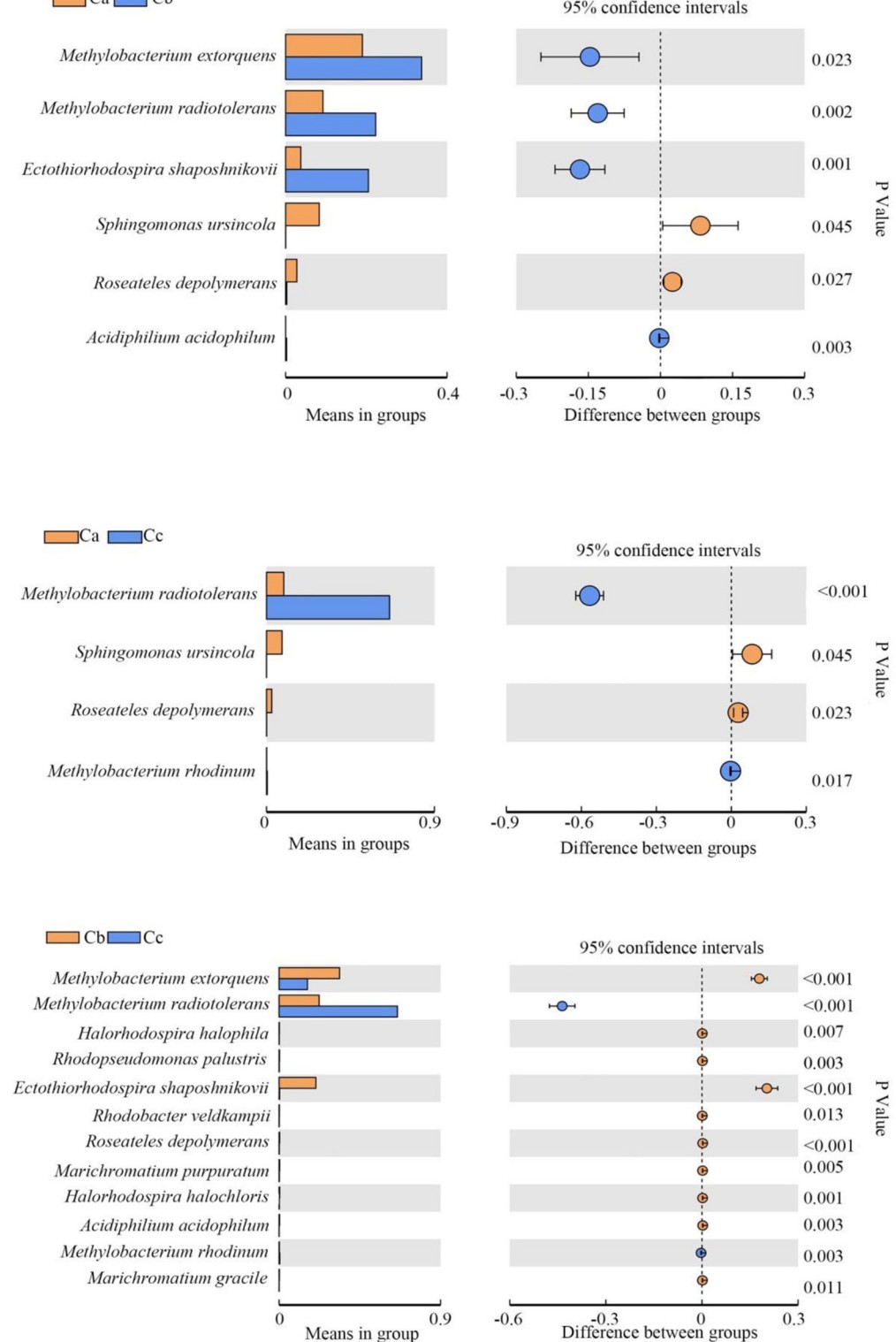

**Fig 5. Photosynthetic bacterial populations showing significant changes among cucumber samples collected at different growth stages.** Ca, cucumber seedling stage; Cb, cucumber flowering stage; Cc, cucumber maturity stage.

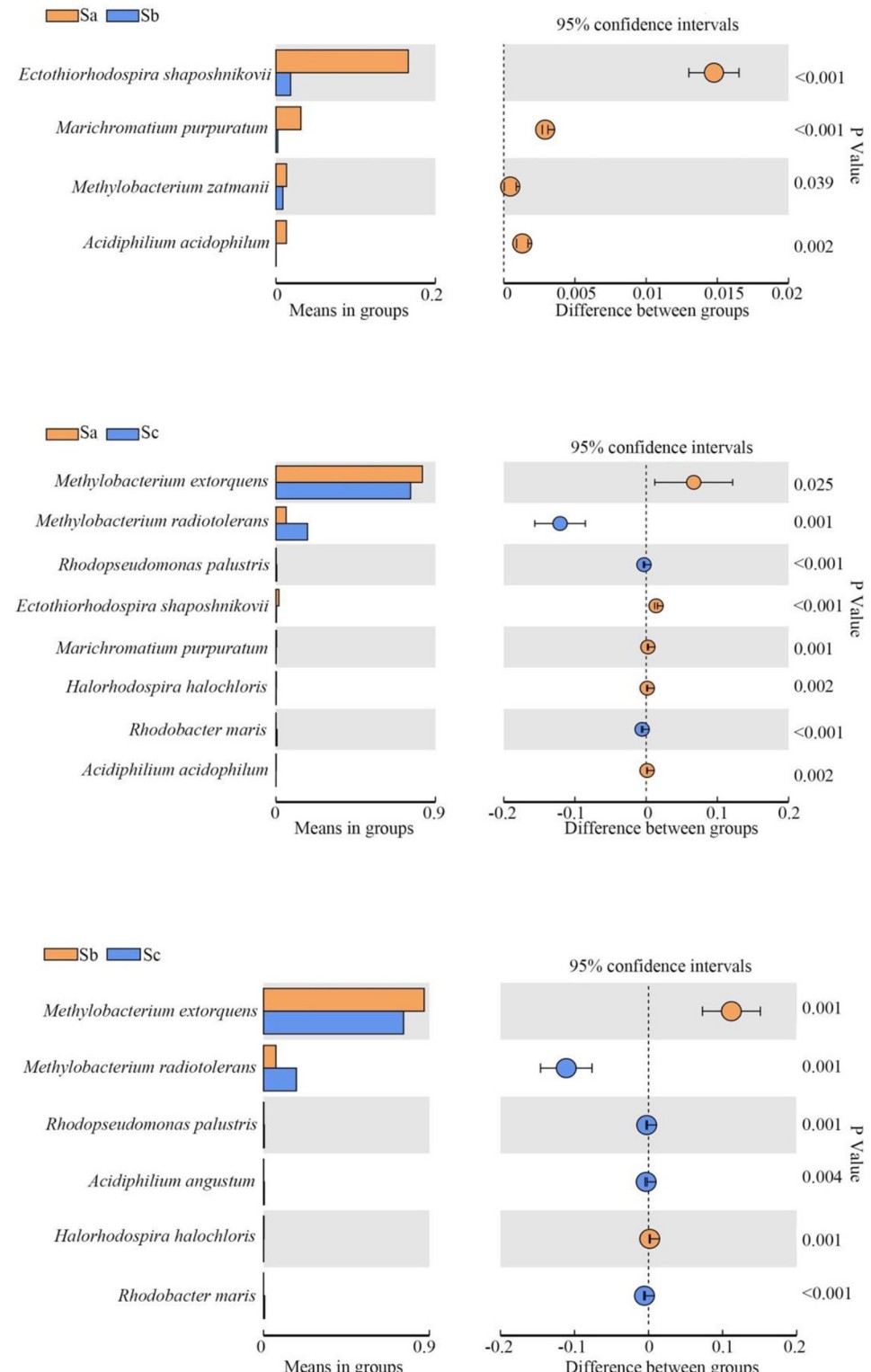

**Fig 6. Photosynthetic bacterial populations showing significant changes among soybean samples collected at different growth stages.** Sa, soybean seedling stage; Sb, soybean flowering stage; Sc, soybean maturity stage.

**Table 3. Photosynthetic bacterial populations showing significant changes among tomato samples collected at three different growth stages.**

| Group | *M. extorquens* | *M. radiotolerans* | *R. depolymerans* | *S. ursincola* |
|---|---|---|---|---|
| Ta | 0.1763±0.0379[b] | 0.0759±0.0084[c] | 0.0116±0.0017[a] | 0.0526±0.013[a] |
| Tb | 0.3026±0.0058[a] | 0.2119±0.0095[a] | 0.0036±0.0005[b] | 0.0003±0.0003[b] |
| Tc | 0.267±0.0129[a] | 0.1646±0.0176[b] | 0.0017±0.0002[b] | 0±0[b] |

The data were analyzed using one-way ANOVA followed by Duncan's multiple range test ($P<0.05$). [a,b,c] Groups with different superscript letters are significantly different ($P<0.05$).

Ta, tomato seedling stage; Tb, tomato flowering stage; Tc, tomato maturity stage.

periods. Abundant phototrophy in the phyllosphere indicated that many microorganisms can fix carbon and must obtain carbon and energy from the oxidation of organic compounds.

*Methylobacterium* was the most abundant bacteria in the phyllosphere of tomato, cucumber, and soybean during their respective growing seasons. *Methylobacterium* has been identified in many crop species [51] and can promote plant growth and development [52]. Knief et al. [53] reported that the distribution of *Methylobacterium* is not plant-specific, supporting the present study. *M. extorquens* and *M. radiotolerans* were found in the photosynthetic bacterial community at different crop growth periods. *M. extorquens* was reported to promote the germination of seeds and plant growth [25, 54]. *M. radiotolerans* was shown to be able to degrade 2,2-bis-(p-chlorophenyl)-1,1-dichloroethylene (DDE) and promote plant growth [55]. Thus, these two bacteria might play an important role in the growth and development of the three plants. *Sphingomonas* differed significantly between the tomato and cucumber growth periods, and the relative abundance at the seedling stage was much higher than in the flowering and mature periods. *Sphingomonas* have unique abilities to degrade refractory contaminants, serve as bacterial antagonists to phytopathogenic fungi, and secrete valuable exopolysaccharides [56]. This study suggests that *Sphingomonas* might play a positive role in the tomato and cucumber seedling periods. These observations are also supported by various studies showing that Sphingomonas promote plant growth [57, 58] and protect against pathogens [24].

Several of the photosynthetic bacteria observed in this study have also been reported as having industrial applications. For example, *R. depolymerans* was originally identified as a degrader of poly-hexamethylene carbonate (PHC), and it was later found to have the ability to use not only PHC but also some other biodegradable plastics [59]. *E. shaposhnikovii* is a phototrophic sulfur bacterium belonging to the Gram-negative subgroup of purple bacteria [60], and it was shown to be able to remove $H_2S$ from wastewater, industrial gases, and biogases [61]. Most of the research on *H. halophila* focused on the photoactive yellow protein [62, 63], but the interactions of these bacteria with plants have not been studied. The role of these

**Table 4. Photosynthetic bacterial populations showing significant changes among the cucumber samples collected at the three different growth stages.**

| Group | *E. shaposhnikovii* | *H. halophila* | *M. extorquens* | *M. radiotolerans* | *R. veldkampii* | *R. palustris* | *R. depolymerans* | *S. ursincola* |
|---|---|---|---|---|---|---|---|---|
| Ca | 0.0376±0.015[b] | 0.285±0.0933[a] | 0.1902±0.0257[b] | 0.0925±0.0157[c] | 0.0332±0.0248[a] | 0.0382±0.0276[a] | 0.0278±0.0042[a] | 0.0833±0.0182[a] |
| Cb | 0.2051±0.0106[a] | 0.0014±0.0002[b] | 0.3369±0.0067[a] | 0.2231±0.0132[b] | 0.0014±0.0002[a] | 0.0021±0.0002[a] | 0.0028±0.0002[b] | 0.0002±0.0001[b] |
| Cc | 0.0025±0.0004[c] | 0.0004±0.0001[b] | 0.158±0.0076[b] | 0.6594±0.0082[a] | 0.0005±0.0002[a] | 0.0007±0.0002[a] | 0.0006±0.0001[b] | 0.0001±0[b] |

The data were analyzed using one-way ANOVA followed by Duncan's multiple range test ($P<0.05$). [a,b,c] Groups with different superscript letters are significantly different ($P<0.05$).

Ca, cucumber seedling stage; Cb, cucumber flowering stage; Cc, cucumber maturity stage.

**Table 5.  Photosynthetic bacterial populations showing significant changes among soybean samples collected at three different growth stages.**

| Group | *E. shaposhnikovii* | *M. extorquens* | *M. radiotolerans* |
|---|---|---|---|
| Sa | 0.0166±0.0006[a] | 0.8267±0.0177[b] | 0.0574±0.0032[b] |
| Sb | 0.0018±0.0001[b] | 0.8718±0.0086[a] | 0.0671±0.0051[b] |
| Sc | 0.0027±0.0005[b] | 0.7599±0.0128[c] | 0.1783±0.0116[a] |

The data were analyzed using one-way ANOVA followed by Duncan's multiple range test (P<0.05). [a,b,c] Groups with different superscript letters are significantly different (P<0.05).

Sa, soybean seedling stage; Sb, soybean flowering stage; Sc, soybean maturity stage.

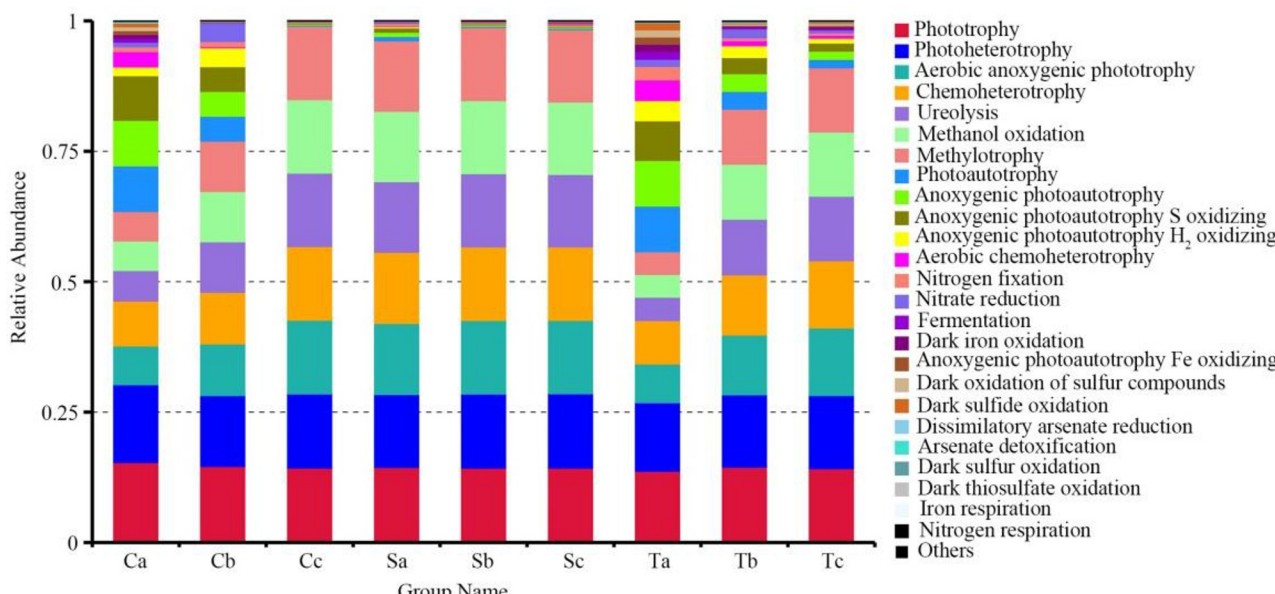

**Fig 7.  Abundance of phyllosphere photosynthetic bacterial function groups predicted with the FAPROTAX tool relative to different groups.**

strains in plant growth and disease resistance will be interesting. On the other hand, the effects of different photosynthetic bacterial strains on plant growth and development need to be further investigated.

## Conclusion

This study provides ecological insights into the photosynthetic bacterial communities in the tomato, cucumber, and soybean phyllosphere at different growth stages. We have demonstrated that photosynthetic bacterial communities are dynamic and change with different growth stages. Still, the molecular mechanisms controlling the changes in photosynthetic bacterial community composition during different plant growth stages require further investigation.

## Supporting information

**S1 Table. Quality filtering of the 35 samples.**
(DOCX)

## Acknowledgments

The authors would like to thank all study participants who were enrolled in this study.

## Author Contributions

**Conceptualization:** Li-Min Zheng, Li-Jie Chen.

**Data curation:** Zhan-Hong Zhang, Xiao-Hua Du, Xiao-Ting Kong.

**Formal analysis:** Li-Min Zheng, Yong Liu.

**Funding acquisition:** De-Yong Zhang.

**Investigation:** Zhong-Yong Wang, Xiao-Hua Du.

**Methodology:** Ju-E Cheng, Pin Su, Zhong-Ying Zhai.

**Project administration:** Jian-Ping Dai.

**Software:** Muhammad Rizwan Hamid.

**Supervision:** Zhan-Hong Zhang, Jian-Ping Dai.

**Validation:** Zhong-Ying Zhai, Xiao-Ting Kong, Yong Liu.

**Visualization:** Li-Min Zheng, Zhong-Yong Wang, Muhammad Rizwan Hamid, Li-Jie Chen.

**Writing – original draft:** Ju-E Cheng, Pin Su.

**Writing – review & editing:** De-Yong Zhang.

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
