## [Decision Letter · Decision Letter 0]

10 Sep 2021

PONE-D-21-13677Dynamical conversion of photosynthetic bacterial communities in tomato, cucumber, and soybean fields over different growth periods characterized by high-throughput sequencingPLOS ONE

Dear Dr. Zhang,

Thank you for submitting your manuscript to PLOS ONE. After careful consideration, we feel that it has merit but does not fully meet PLOS ONE’s publication criteria as it currently stands. Therefore, we invite you to submit a revised version of the manuscript that addresses the points raised during the review process.

We look forward to receiving your revised manuscript.

Kind regards,

Suzanne L. Ishaq, PhD

Academic Editor

PLOS ONE

Journal Requirements:

3. We note that you are reporting an analysis of a microarray, next-generation sequencing, or deep sequencing data set. PLOS requires that authors comply with field-specific standards for preparation, recording, and deposition of data in repositories appropriate to their field. Please upload these data to a stable, public repository (such as ArrayExpress, Gene Expression Omnibus (GEO), DNA Data Bank of Japan (DDBJ), NCBI GenBank, NCBI Sequence Read Archive, or EMBL Nucleotide Sequence Database (ENA)). In your revised cover letter, please provide the relevant accession numbers that may be used to access these data. For a full list of recommended repositories, see http://journals.plos.org/plosone/s/data-availability#loc-omics or http://journals.plos.org/plosone/s/data-availability#loc-sequencing

Reviewers' comments:

Reviewer's Responses to Questions

**Comments to the Author**

1. Is the manuscript technically sound, and do the data support the conclusions?

Reviewer #1: Yes

Reviewer #2: Yes

2. Has the statistical analysis been performed appropriately and rigorously? 

Reviewer #1: Yes

Reviewer #2: Yes

3. Have the authors made all data underlying the findings in their manuscript fully available?

Reviewer #1: Yes

Reviewer #2: Yes

4. Is the manuscript presented in an intelligible fashion and written in standard English?

Reviewer #1: Yes

Reviewer #2: No

5. Review Comments to the Author

Reviewer #1: 1. Summary: Photosynthetic bacteria are an important component of the phyllosphere, as they play roles in plant health, plant function, and disease suppression. Though there are potential applications for agricultural production, many photosynthetic bacterial communities in the phyllosphere have not been characterized. The authors hypothesize that photosynthetic bacteria are common in the phyllosphere and that the growth period of crops will affect community composition. The results are in support of this hypothesis, as 3,150,689 quality-filtered reads were found in 6725 identified OTUs across 35 pooled samples, alpha diversity changed throughout the growth stages of tomato, cucumber, and soybean, and community dissimilarity was affected by plant growth stage. The reviewer finds the manuscript to be technically sound and that the data supports the authors' conclusions.

2. Figures, tables, and captions: The figures are clear and easy to understand, as are the captions.

4. There are no issues with the writing that would detract from understanding the work, save for one instance where clarity is needed.

Lines 50 - 51: This sentence could be reworded; it reads like it is missing a phrase. As written, this sentence implies that photosynthetic bacteria have only been in the phyllosphere as of recently. Is the sentence meant to convey that the existence of photosynthetic bacteria in the phyllosphere was recently confirmed?

Line 54: The phrase "trigger inducible" seems redundant. For example, the literature cited uses the phrase "induces systemic resistance" in the title.

Reviewer #2: The manuscript by Ju-E et al. provides relevant information on changes in bacterial communities within different crop field caused by the different growth periods. However, substantial revision is needed to improve the clarity of the information provided, particularly regarding to the materials and methods, results, and discussion. Additionally, the manuscript should be proofread to improve the quality of the writing. More specific comments are provided below. Please proofread the manuscript to eliminate typos, misplaced words, missing spaces, sentences that need to be restructured to avoid confusion and improve coherence, and formatting errors.

The title of the manuscript is not very clear, it should be like “Metagenomic analysis of dynamical conversion of photosynthetic bacterial communities in different crop fields over different growth periods”.

Page 2, line 33. Please consider to remove “and” from sentence. different 33 crop species and at different growth stages.

Page 2, line 38. Consider to remove “and” among plant growth and development.

Page 3, line 49. “proteobacteria” Be consistent with the use of bacterial name in italic and with the use of capital letters. Check throughout the manuscript.

Page 4, line 82. Add “help to identify” instead of “help reveal”.

Page 5, line 88 and 89. These two sentences are useless in the start, adjust it in the last before DNA isolation and consider it like “a total of 35 samples were collected from leaves of three crops at different growth stages for subsequent microbe isolation and high throughput sequencing process.

Page 5, line 95. The methodology part is lack of crop plantation points, in which base the block were randomly selected? Fertilization/nutrients rate was same in every block? How many seed were cultivated in every block? Was the seed quantity was varied or same per block? After how much time the samples were collected?

Page 5, line 104. This sentence is unclear, please consider to rewrite it.

Page 8, line 151. Consider to mention the quality filtration information of all the 35 samples as a supplementary table.

Page 11, line 223-226. Fig 3C doesn’t representing the mentioned bacterium except few one.

Page 11, line 227. Explain all the significantly diverse bacterial species among different crop at different growth stages.

Page 15, line 269. Table 5, Consider to mentioned all the a, b, c (P-value) along with the tables. Check with every table showing statistical data.

Page 17, line 294-296. The sentence should be like, the results showed that the photosynthetic bacterial community was common in the phyllosphere and the composition of photosynthetic bacterial community at the phylum and genus levels were similar among the growth stages of the three crops.

6. PLOS authors have the option to publish the peer review history of their article (what does this mean?). If published, this will include your full peer review and any attached files.

Reviewer #1: No

Reviewer #2: No

---

## [Author Response · Author response to Decision Letter 0]

15 Nov 2021

Manuscript ID: PONE-D-21-13677

Title: Dynamical conversion of photosynthetic bacterial communities in tomato, cucumber, and soybean fields over different growth periods characterized by high-throughput sequencing

Journal: PLOS ONE

Response to Reviewers' comments

Dear Editor, 

 We thank you for your careful consideration of our manuscript. We appreciate your response and overall positive initial feedback and made modifications to improve the manuscript. After carefully reviewing the comments made by the Reviewers, we have modified the manuscript to improve the presentation of our results and their discussion, therefore providing a complete context for the research that may be of interest to your readers.

 We hope that you will find the revised paper suitable for publication, and we look forward to contributing to your journal. Please do not hesitate to contact us with other questions or concerns regarding the manuscript.

Best regards,

 

Journal Requirements

 Response: We thank the Journal. The manuscript was revised accordingly.

 Response: It was corrected.

3. We note that you are reporting an analysis of a microarray, next-generation sequencing, or deep sequencing data set. PLOS requires that authors comply with field-specific standards for preparation, recording, and deposition of data in repositories appropriate to their field. Please upload these data to a stable, public repository (such as ArrayExpress, Gene Expression Omnibus (GEO), DNA Data Bank of Japan (DDBJ), NCBI GenBank, NCBI Sequence Read Archive, or EMBL Nucleotide Sequence Database (ENA)). In your revised cover letter, please provide the relevant accession numbers that may be used to access these data. For a full list of recommended repositories, see http://journals.plos.org/plosone/s/data-availability#loc-omics or http://journals.plos.org/plosone/s/data-availability#loc-sequencing

 Response: We thank the Journal. In this study, all fastq files are available from the NCBI database (bacterial sequences SRA: SRP193589; BioProject: PRJNA533201; BioSample: SAMN11444424-SAMN11444390; URL: https://www.ncbi.nlm.nih.gov/bioproject/533201).

 Response: We provided the ORCID numbers.

 Response: The reference list was verified.

 

Reviewer #1

1. Summary: Photosynthetic bacteria are an important component of the phyllosphere, as they play roles in plant health, plant function, and disease suppression. Though there are potential applications for agricultural production, many photosynthetic bacterial communities in the phyllosphere have not been characterized. The authors hypothesize that photosynthetic bacteria are common in the phyllosphere and that the growth period of crops will affect community composition. The results are in support of this hypothesis, as 3,150,689 quality-filtered reads were found in 6725 identified OTUs across 35 pooled samples, alpha diversity changed throughout the growth stages of tomato, cucumber, and soybean, and community dissimilarity was affected by plant growth stage. The reviewer finds the manuscript to be technically sound and that the data supports the authors' conclusions.

2. Figures, tables, and captions: The figures are clear and easy to understand, as are the captions.

 Response: We thank the Reviewer for taking the time to review our work and for the comments.

4. There are no issues with the writing that would detract from understanding the work, save for one instance where clarity is needed.

Lines 50 - 51: This sentence could be reworded; it reads like it is missing a phrase. As written, this sentence implies that photosynthetic bacteria have only been in the phyllosphere as of recently. Is the sentence meant to convey that the existence of photosynthetic bacteria in the phyllosphere was recently confirmed?

Line 54: The phrase "trigger inducible" seems redundant. For example, the literature cited uses the phrase "induces systemic resistance" in the title.

 Response: We thank the Reviewer. The entire manuscript was proofread, including the two statements mentioned above.

 

Reviewer #2

The title of the manuscript is not very clear, it should be like “Metagenomic analysis of dynamical conversion of photosynthetic bacterial communities in different crop fields over different growth periods”.

 Response: We thank the Reviewer for the comment. The title was revised as suggested.

Page 2, line 33. Please consider to remove “and” from sentence. different 33 crop species and at different growth stages.

 Response: It was corrected.

Page 2, line 38. Consider to remove “and” among plant growth and development.

 Response: It was corrected.

Page 3, line 49. “proteobacteria” Be consistent with the use of bacterial name in italic and with the use of capital letters. Check throughout the manuscript.

 Response: It was corrected. The names were verified.

Page 4, line 82. Add “help to identify” instead of “help reveal”.

 Response: It was corrected.

Page 5, line 88 and 89. These two sentences are useless in the start, adjust it in the last before DNA isolation and consider it like “a total of 35 samples were collected from leaves of three crops at different growth stages for subsequent microbe isolation and high throughput sequencing process.

 Response: It was corrected as suggested.

Page 5, line 95. The methodology part is lack of crop plantation points, in which base the block were randomly selected? Fertilization/nutrients rate was same in every block? How many seed were cultivated in every block? Was the seed quantity was varied or same per block? After how much time the samples were collected?

 Response: We thank the Reviewer for the comment. The experiment was conducted from May to August 2018 at the experimental farm of the Hunan Academy of Agricultural Sciences, Changsha, Hunan Province, China (28.22oN, 113.26oE). Tomato cv. Zuan-hong-meili (XinShu Seed Co., Ltd., Beijing, China), cucumber cv. Shuyan 10 (XinShu Seed Co., Ltd., Beijing, China), and soybean cv. Su-qing III (Nanjing Ideal Seedings Co., Ltd., Nanjing, China) were selected for the study. A large block was randomly picked from the experimental farm. Then it was divided into nine small blocks. The size of each block was approximately 40-50 m2, and the nutrients were consistent. Then, 300 plants were planted in each block of cucumber, tomato, and soybean. Each crop was planted in three randomly selected blocks on May 15, 2018 and sampled at the seedling (June 15, 2018), flowering (July 15, 2018), and mature (August 25, 2018) growth stages. This information was added to the Methods.

Page 5, line 104. This sentence is unclear, please consider to rewrite it.

 Response: The sentence was revised.

Page 8, line 151. Consider to mention the quality filtration information of all the 35 samples as a supplementary table.

 Response: We thank the Reviewer. We now present the quality filtration process in Supplementary Table S1.

Page 11, line 223-226. Fig 3C doesn’t representing the mentioned bacterium except few one.

Response: We thank the Reviewer. We revised the description.

Page 11, line 227. Explain all the significantly diverse bacterial species among different crop at different growth stages.

 Response: We thank the Reviewer. We revised the description.

Page 15, line 269. Table 5, Consider to mentioned all the a, b, c (P-value) along with the tables. Check with every table showing statistical data.

 Response: We thank the Reviewer. We verified all tables. Groups with different superscript letters are significantly different (P<0.05).

Page 17, line 294-296. The sentence should be like, the results showed that the photosynthetic bacterial community was common in the phyllosphere and the composition of photosynthetic bacterial community at the phylum and genus levels were similar among the growth stages of the three crops.

 Response: We thank the Reviewer. The sentence was revised as suggested.

---

## [Decision Letter · Decision Letter 1]

28 Dec 2021

Metagenomic analysis of the dynamical conversion of photosynthetic bacterial communities in different crop fields over different growth periods

PONE-D-21-13677R1

Dear Dr. Zhang,

We’re pleased to inform you that your manuscript has been judged scientifically suitable for publication and will be formally accepted for publication once it meets all outstanding technical requirements.

Kind regards,

Suzanne L. Ishaq, PhD

Academic Editor

PLOS ONE

Additional Editor Comments (optional):

Reviewers' comments:

Reviewer's Responses to Questions

**Comments to the Author**

1. If the authors have adequately addressed your comments raised in a previous round of review and you feel that this manuscript is now acceptable for publication, you may indicate that here to bypass the “Comments to the Author” section, enter your conflict of interest statement in the “Confidential to Editor” section, and submit your "Accept" recommendation.

Reviewer #1: All comments have been addressed

Reviewer #2: All comments have been addressed

2. Is the manuscript technically sound, and do the data support the conclusions?

Reviewer #1: Yes

Reviewer #2: Yes

3. Has the statistical analysis been performed appropriately and rigorously? 

Reviewer #1: Yes

Reviewer #2: Yes

4. Have the authors made all data underlying the findings in their manuscript fully available?

Reviewer #1: Yes

Reviewer #2: Yes

5. Is the manuscript presented in an intelligible fashion and written in standard English?

Reviewer #1: Yes

Reviewer #2: Yes

6. Review Comments to the Author

Reviewer #1: All previous comments were addressed. However, the additional information provided in the Materials and Methods section does raise additional questions about nutrient application rates and the experimental design.

Materials and methods:

Lines 101-105: More of the experimental design has been added, but the usage of "block" for two different levels is somewhat confusing. Replacing "block" where it appears in Line 101 (referring to the field picked for experimentation) with another term may be helpful.

Line 103: For the sake of experimental replication, the information on nutrient application could be included as supplemental information or be briefly described beyond "The nutrients were consistent."

Lines 103-104: Having this sentence come before the one detailing random assignment of crops to blocks is confusing. It is also not clear that each block is planted in a monoculture. It could help to combine sentences and clarify that each crop was randomly assigned to three replicate blocks and planted in a monoculture of 300 seedlings. Or, a simple diagram could be included.

The manuscript could also benefit from further proofreading and editing for clarity. The following comments are some suggestions along those lines.

Abstract:

Lines 27 - 28: As the sentence is currently written, the apostrophe is no longer needed. Alternatively, the phrase "the knowledge...in field crops" could be replaced with "knowledge of photosynthetic bacterial community dynamics in field crops" to make the first sentence of the abstract more concise.

Line 31: Consider restructuring this sentence to place the growth stages closer to the list of crops being sampled (ex: "...collected from the seedling, flowering, and mature stages of tomato, cucumber, and soybean plants.")

Line 32: Consider replacing "they" with the subject (samples) for clarity

Introduction:

Line 70: Consider moving the verb in this sentence and changing tense. As currently written, the sentence may imply the bacteria no longer have the properties described. (ex: "Methylobacterium spp., the first...bacteria, were found to have..."

Discussion:

In the second sentence, it would help to clarify that these are your results and not results from the aforementioned studies.

Figures:

The color palette may be difficult for people with color deficiencies to view.

Reviewer #2: (No Response)

7. PLOS authors have the option to publish the peer review history of their article (what does this mean?). If published, this will include your full peer review and any attached files.

Reviewer #1: No

Reviewer #2: No

---

## [Editor Report · Acceptance letter]

6 Jul 2022

PONE-D-21-13677R1 

Metagenomic analysis of the dynamical conversion of photosynthetic bacterial communities in different crop fields over different growth periods 

Dear Dr. Zhang:

I'm pleased to inform you that your manuscript has been deemed suitable for publication in PLOS ONE. Congratulations! Your manuscript is now with our production department. 

Kind regards, 

on behalf of

Dr. Suzanne L. Ishaq 

Academic Editor

PLOS ONE